# Test-retest reliability of the HEXACO-100— And the value of multiple measurements for assessing reliability

**Sam Henry** [1]☉*, **Isabel Thielmann** [2‡], **Tom Booth** [1‡], **René Mõttus** [1,3☉]

**1** Department of Psychology, University of Edinburgh, Edinburgh, Midlothian, United Kingdom, **2** Department of Psychology, University of Koblenz-Landau, Koblenz, Landau, Germany, **3** Institute of Psychology, University of Tartu, Tartu, Estonia

☉ These authors contributed equally to this work.
‡ IT and TB also contributed equally to this work.
* samuel.henry@ed.ac.uk

**Data Availability Statement:** All test-retest data files will be available from the project page on the Open Science Foundation database (https://osf.io/wz3du/) and are also included in this submission. However, the cross-rater agreement data included

## Abstract

Despite the widespread use of the HEXACO model as a descriptive taxonomy of personality traits, there remains limited information on the test-retest reliability of its commonly-used inventories. Studies typically report internal consistency estimates, such as alpha or omega, but there are good reasons to believe that these do not accurately assess reliability. We report 13-day test-retest correlations of the 100- and 60-item English HEXACO Personality Inventory-Revised (HEXACO-100 and HEXACO-60) domains, facets, and items. In order to test the validity of test-retest reliability, we then compare these estimates to correlations between self- and informant-reports (i.e., cross-rater agreement), a widely-used validity criterion. Median estimates of test-retest reliability were .88, .81, and .65 ($N$ = 416) for domains, facets, and items, respectively. Facets' and items' test-retest reliabilities were highly correlated with their cross-rater agreement estimates, whereas internal consistencies were not. Overall, the HEXACO Personality Inventory-Revised demonstrates test-retest reliability similar to other contemporary measures. We recommend that short-term retest reliability should be routinely calculated to assess reliability.

## Introduction

The HEXACO Personality Inventory-Revised (HEXACO-PI-R [1]) is currently one of the most widely used personality questionnaires in psychology and beyond. Several key properties of its domain and facet scales such as convergent and discriminant validity, gender differences, and measurement invariance across various countries and translations have been reported in recent large-sample studies [2, 3]. The scales also demonstrate associations with numerous life outcomes [4–6]. Surprisingly, however, one of the most basic of its psychometric properties, test-retest reliability (or retest reliability; $r_{TT}$), has rarely been assessed, and never in sufficiently large samples [7–9].

Reliability denotes "the consistency of a measure with itself" [10; p.464] or of two independent assessments of the same inventory [11]. Whereas internal reliability or internal

in the additional analyses was not collected by the authors of this manuscript. The analyses of facet alphas and cross-rater agreement estimates were simply copied from Table 4 in Lee & Ashton (2018) and can thus be accessed in the published article itself. We can confirm that we did not have any special access privileges to the single-item data. Though the authors of Lee & Ashton (2018) do not specifically indicate in their article how to access these data, we simply sent them an email requesting the raw self-/observer data and explaining how we intended to use them. We would thus confirm that, to our knowledge, others can access these datasets and would be able to access these data in the same manner as the authors of the present manuscript by contacting the authors Lee & Ashton (2018) in a similar manner.

**Funding:** The corresponding author (SH) was supported by two Research Support Grants (grant numbers unspecified as the funds come from a common pool used by research students) from the University of Edinburgh School of Philosophy, Psychology, and Language Sciences. Information on these grants may be found at https://www.ed.ac.uk/ppls/linguistics-and-english-language/current/postgraduate/fees-and-funding/funding-phd-research. The funders had no role in study design, data collection and analysis, decision to publish, or preparation of the manuscript.

**Competing interests:** The authors have declared that no competing interests exist.

consistency represents the consistency of responses for a trait using non-identical parallel forms, short-term $r_{TT}$ quantifies the extent to which test takers agree with themselves on the *exact same items* being measured at different time points and hence independently, assuming that test takers do not remember their previous responses and copy these at second testing. For many, $r_{TT}$ *is* reliability, whereas internal consistency estimates like alpha ($\alpha$) and omega ($\omega$) are seen to have only been designed to imperfectly approximate reliability when $r_{TT}$ is not available [12]. As succinctly noted by McCrae and colleagues [13]: "[i]nternal consistency of scales can be useful as a check on data quality but appears to be of limited utility for evaluating the potential validity of developed scales, and it should not be used as a substitute for retest reliability" (p. 28). In many cases, high internal consistency, reflecting redundancy among test constituents, is even *un*desirable because it may mean poor coverage of the content of the construct being measured–especially when constructs are broad–or an impractically long test. We note that $\alpha$ does not measure internal consistency of items *per se*: with hundreds or thousands of items, any scale has high $\alpha$ but can have practically zero consistency among individual items. Instead, a scale's $\alpha$ indexes the expected consistency among hypothetical item aggregates containing the same number of items as the scale. Nonetheless, in line with common usage and to clearly distinguish it from $r_{TT}$, we will also refer to $\alpha$ as a measure of internal consistency.

In addition to its tighter theoretical adherence to reliability, $r_{TT}$ has empirical advantages when compared to internal consistency. Because internal reliability confounds unreliability with unique trait information in items [14], it tends to be lower than $r_{TT}$ [15], and this difference may even be underestimated due to occasion-specific state effects inflating internal consistency. Even more importantly, $r_{TT}$, but not internal consistency, is a good predictor of scales' cross-rater agreement ($r_{CA}$) and long-term stability, two of the most straightforward and broadly applicable psychometric validity criteria [13, 16]. These associations with validity criteria follow with $r_{TT}$ being a more accurate reliability estimate than internal consistency, as reliability necessarily provides an upper bound for validity. Unlike internal reliability, $r_{TT}$ can also be estimated for individual items. This provides researchers an empirical criterion to assess item quality–where items with low $r_{TT}$ either index characteristics on which respondents cannot even agree with themselves or are simply ambiguous. Although some items may assess a specific pattern of thought, feeling, or behavior that genuinely changes over a brief time span, we contend that traits, even those assessed by single items, should be largely stable over the brief time periods typically used to measure $r_{TT}$, and those items with particularly low $r_{TT}$s ought to be replaced with more reliable ones where possible.

While indices of internal consistency (most prominently $\alpha$) have been widely reported for HEXACO measures [2], only a handful of studies have reported $r_{TT}$ estimates for them. For example, previous work [9] assessed seven-month $r_{TT}$ of the full 200-item HEXACO-PI-R to compare it with a variety of other item properties and found a mean single-item $r_{TT}$ = .56 ($N$ = 188; $SD$ = .10; range = .31 to .80), with mean domain $r_{TT}$ = .85 (range = .79 to .90). Others [7] found similar results for seven-month $r_{TT}$ of HEXACO domains using the 60-item version of the HEXACO-PI-R (HEXACO-60 [17]), albeit with a very small sample ($N$ = 31; $M$ = .81; range = .72 to .90). More recently, 2-year $r_{TT}$ ($N$ = 214) estimates were reported [8] for only the Honesty-Humility, Agreeableness, and Conscientiousness domains and facets of the 100-item HEXACO-PI-R (HEXACO-100 [18]). Domain reliabilities were $r_{TT}$ = .80, .75, and .74, respectively, and median facet reliability was $r_{TT}$ = .69 (range = .58 to .78).

Despite providing good preliminary evidence for the temporal stability of HEXACO domains, all three studies used a testing interval more appropriate for long-term stability estimates, rather than for quantifying $r_{TT}$ [19]. Longer retest intervals over many months or even years confound unreliability (or "transient error") with actual personality change, while short-

term intervals such as two weeks [20, 21] or less [22] are more appropriate to quantify $r_{TT}$ because true change is unlikely to occur, but test takers are also unlikely to retrieve their previous responses from memory. Cattell et al. [23] recommended one or two months as the ideal interval where no true change should occur and "memory effects" [24] are still unlikely. However, more recent research [20] suggests that even shorter timespans still do not evoke memory effects, finding no discernible difference in Big Five (i.e., NEO) personality domain $r_{TT}$s measured two months versus two weeks apart. For single items, Kosinski et al. [25] found that median $r_{TT}$s declined almost linearly from one-day to one-, two-, three-, and four-week retest intervals, though the range was relatively narrow ($r_{TT}$s = .63 to .70) and did not include a two-month interval for further comparison. Thus, while the exact interval that distinguishes between unreliability and true change is unclear (and perhaps impossible to resolve), we suggest–in line with previous research [e.g., 16, 20, 26]–that two weeks strikes a balance between mitigating both the possibility of true change and the likelihood that participants recall and repeat their responses from the first measurement.

Given the popularity of HEXACO-PI-R scales and that $r_{TT}$ is likely a superior measure of the scales' most fundamental property, reliability, a comprehensive study of HEXACO-PI-R domain, facet, and item $r_{TT}$s over a brief time period should be of wide interest to both personality researchers and those applying HEXACO-PI-R scales in other (including practical) settings. Thus, we report those estimates here for the English HEXACO-100 and HEXACO-60 (where the latter uses a subset of items from the former); although the longer 200-item version exists, the HEXACO-100 and HEXACO-60 are much more commonly used (in a meta-analysis [2], the two shorter scales were used in 443 of 489 studies that administered a HEXACO-PI-R inventory). To evaluate the criterion validity of $r_{TT}$ versus internal consistency estimates, we also explore the extent to which these indices correlate with facet-scale and (where possible) single-item $r_{CA}$s [18]. Thus, similarly to previous work [13], we expected $r_{TT}$ but not internal consistency estimates to correlate with $r_{CA}$s at all levels of investigation. Finally, we report the $r_{TT}$s for individual HEXACO-PI-R items, which could suggest those in need of replacement in future iterations of the inventory and inform research on the properties of (un)reliable items.

## Material and methods

### Participants

Participants were recruited via Prolific Academic from a cohort of participants ($N$ = 639) that forms part of an ongoing project led by the first and last author on this manuscript. These participants had previously provided survey responses for personality, life outcomes, and item properties. All participants provided informed, written consent online in all previous waves of data collection, which was approved by the University of Edinburgh School of Philosophy, Psychology, and Language Sciences Research Ethics Committee (Refs 123-1920/1 and 123-1920/2). The present study, submitted as a new iteration of the previous one, received approval from the same committee on 11 September 2020 (Ref 400-1920/3).

We invited these participants to complete the English HEXACO-100 [18] twice in two weeks. At the first administration (T1), participants gave written online consent before being directed to the survey. T1 was released at 11:55 GMT on 23 September 2020 and closed at 19:01 GMT on 28 September after achieving the planned sample size ($N$ = 450). The second survey (T2) was released in two waves to account for variance in T1 start dates and restrict the range of retest intervals. Specifically, we published the first T2 survey at 11:15 GMT on 6 October for participants who completed T1 on 23–25 September ($n$ = 421), then published the second wave two days later (12:11 GMT on 8 October) for participants who completed T1 between 26 and 28 September. Ultimately, 423 participants (48.5% female; $M$ age = 26.9, $SD$

age = 7.9, age range = 19 to 69) completed both T1 and T2 assessments. Pending a successful quality control check, all participants received a total compensation of £2.00 for participation at both time points.

Following recommendations from a similar study [27], we excluded participants whose profile consistency ($q$, calculated as the overall correlation between responses across all items at each measurement occasion) was exceptionally low–they used a cut-off of $q = 0.25$ for consistency estimates taken from repeated measures in the same session. Given our considerably longer testing interval, we were more lenient and only removed participants whose profile consistency was three or more standard deviations below the median $q = 0.66$ (i.e., $q \leq 0.12$). This excluded seven participants, leaving a final sample of $N = 416$ (49.0% female; $M$ age = 26.9, $SD$ age = 7.9, age range = 19 to 69). For a hypothesized mean $r_{TT}$ of .65 (based on previous research [16, 25]), this final sample size entailed an average predicted standard error ($SE$) of .037.

With respect to other demographic variables, our sample was rather heterogeneous. No one country of birth exceeded 100 participants, with the largest representation coming from Portugal ($n = 84$), United Kingdom ($n = 76$), Poland ($n = 49$), Italy ($n = 33$), Greece ($n = 30$), Spain ($n = 22$), and the United States ($n = 15$). All other nations had $n$s < 10. English was underrepresented as a first language, with just under a quarter of the sample being native English speakers ($n = 94$); first language mapped fairly consistently with country of birth, although there was a greater variety of the latter (50 countries of birth versus 33 first languages).

Finally, about 1/3 ($n = 137$ and 135, respectively) of the sample was missing data for student and employment status, but among the remaining subsamples, about half (53.1%) were students and over two thirds (69.4%) claimed to be employed in some capacity. In summary, although our sample was not representative of any one population, we did manage to capture a relatively wide variety of cultures, ages, and occupational circumstances.

## Measure

The HEXACO-100 measures the six HEXACO domains, their respective four facets (see Table 1 for domain and facet names), and an additional interstitial facet—Altruism—for a total of 25 facets with four items each [18]. The measure also contains all 60 items of the shorter form of the inventory, the HEXACO-60 [17], which contains 10 items for each domain. The HEXACO-60 does not include items for the Altruism facet, and is generally not intended to assess HEXACO facets. Full scales and scoring keys are freely available at https://hexaco.org/hexaco-inventory.

## Analyses

After recoding reverse-keyed items, we calculated domain and facet scale scores for both HEXACO-100 and HEXACO-60 using mean scores of their associated items. Though facet scores are not typically calculated for the latter, we deemed it worthwhile to do so for the purpose of comparing αs and $r_{TT}$s. Using the psych() package in R Version 4.1.1 [28, 29], we calculated internal consistencies (Cronbach's α) for facets and domains (using item scores to estimate domains' αs). Because some researchers [12] recommend omega (ω) as a more appropriate measure for internal reliability than α, we calculated domain and facet ωs for comparison with αs. They correlated .98 and .99, respectively; ω estimates are available in the Online Supplement (https://osf.io/wz3du/). Test-retest reliabilities for domains, facets, and items were all estimated as the correlation between their scores at T1 and T2.

We report α and $r_{TT}$ estimates from our sample alongside the α and $r_{CA}$ estimates from a previous meta-analysis [18]. They reported the estimates from a large sample comprised of

**Table 1. Empirical properties of HEXACO-100 domains and facets.**

| Scale | Our study | | | Lee & Ashton (2018) | | |
|---|---|---|---|---|---|---|
| | α | $r_{TT}$ | Δ | $r_{CA}$ | $α_{Student}$ | $α_{Online}$ |
| H: Honesty-Humility | .88 | .89 | .01 | .46 | .82 | .89 |
| E: Emotionality | .84 | .88 | .04 | .61 | .84 | .84 |
| X: eXtraversion | .89 | .92 | .03 | .56 | .85 | .86 |
| A: Agreeableness | .87 | .86 | (.01) | .47 | .84 | .86 |
| C: Conscientiousness | .87 | .88 | .01 | .52 | .84 | .82 |
| O: Openness to Experience | .83 | .88 | .05 | .56 | .81 | .82 |
| H1: Sincerity | .83 | .75 | (.08) | .20 | .66 | .78 |
| H2: Fairness | .85 | .86 | .01 | .45 | .76 | .83 |
| H3: Greed Avoidance | .83 | .84 | .01 | .47 | .81 | .83 |
| H4: Modesty | .79 | .80 | .01 | .30 | .68 | .79 |
| E1: Fearfulness | .72 | .81 | .09 | .51 | .70 | .70 |
| E2: Anxiety | .75 | .81 | .06 | .40 | .64 | .73 |
| E3: Dependence | .78 | .80 | .02 | .44 | .80 | .76 |
| E4: Sentimentality | .78 | .84 | .06 | .47 | .70 | .73 |
| X1: Social Self-Esteem | .74 | .85 | .11 | .38 | .67 | .70 |
| X2: Social Boldness | .74 | .83 | .09 | .53 | .76 | .72 |
| X3: Sociability | .80 | .85 | .05 | .45 | .71 | .77 |
| X4: Liveliness | .81 | .83 | .02 | .45 | .76 | .78 |
| A1: Forgivingness | .82 | .78 | (.04) | .35 | .74 | .78 |
| A2: Gentleness | .73 | .75 | .02 | .35 | .66 | .72 |
| A3: Flexibility | .70 | .76 | .06 | .35 | .61 | .64 |
| A4: Patience | .82 | .82 | 0 | .43 | .79 | .80 |
| C1: Organization | .74 | .81 | .07 | .52 | .74 | .73 |
| C2: Diligence | .80 | .83 | .03 | .37 | .70 | .71 |
| C3: Perfectionism | .76 | .79 | .03 | .42 | .69 | .69 |
| C4: Prudence | .77 | .74 | (.03) | .33 | .69 | .70 |
| O1: Aesthetic Appreciation | .69 | .83 | .14 | .49 | .66 | .65 |
| O2: Inquisitiveness | .66 | .80 | .14 | .45 | .66 | .70 |
| O3: Creativity | .78 | .87 | .09 | .50 | .75 | .73 |
| O4: Unconventionality | .54 | .78 | .24 | .36 | .52 | .59 |
| Interstitial: Altruism | .66 | .75 | .09 | .36 | .59 | .66 |
| *Domain Median* | *.87* | *.88* | *.01* | *.54* | *.84* | *.85* |
| *Facet Median* | *.77* | *.81* | *.04* | *.43* | *.70* | *.73* |

α = alpha internal consistency. $r_{TT}$ = 13-day test-retest reliability. $r_{CA}$ = cross-rater agreement. Δ = $r_{TT}$—α in the present sample, with instances of α > $r_{TT}$ in parentheses.

self- and informant reports from university students ($N$ = 2,863 pairs), and a very large sample of participants providing only self-reports via a survey link on the HEXACO website ($N$ = 100,318). Full details on the samples, data collection, and results may be found in the original publication [18].

Finally, we report Spearman's correlations between αs, $r_{TT}$s, and $r_{CA}$s of HEXACO-100 facets as well as the correlation between items' $r_{TT}$s and $r_{CA}$s. Items' $r_{CA}$s were not published in the original paper, but the authors kindly made their data available to us, so we calculated these. We opted not to conduct these analyses for domains because we deemed that correlations of vectors of six values would not provide additional informative value. We were

particularly interested in the extent to which facets' $r_{TT}$ and α correlated with $r_{CA}$: in other words, how strongly reliability criteria correlate with a common validity criterion.

Code and data that may be used to reproduce all analyses, as well as a data file containing final outputs, can be found in the Online Supplement at https://osf.io/wz3du/.

# Results

## Completion time, interval length, and language

Median time to complete the survey at T1 was 11' 33", (SD = 8'22", IQR = 8'55" to 16'3"). For T2, we removed times for two participants with extreme values (approximately 25 and 72 hours). The resulting completion times were slightly shorter than T1 (median = 10'44", SD = 9'20", IQR = 8'13" to 13' 20"). On average, participants provided complete data in about 13 days (median T1-T2 interval = 12 days, 23 hours, 37 minutes; SD = 1 day, 6 hours, 41 minutes), with a vast majority completing the survey by the end of two weeks; only $n$ = 56 completed the survey after 14 days and no participants took more than 19 days to complete it.

To evaluate whether interval length moderated $r_{TT}$ estimates, we calculated mean T1-T2 profile correlations for each participant, then correlated these with the length of time in seconds between T1 and T2 start times. The resulting correlation was $r$ = .07 ($p$ = .14), suggesting no relationship between the number of days between administrations and overall reliability. We also examined correlations between T1-T2 profile correlations and time taken to complete the survey at T1, T2, and the average of these, but found no evidence that delay between measurements affected participants' overall consistency over time either: $r$s = -.08 ($p$ = .12), .01 ($p$ = .84), and -.04 ($p$ = .42), respectively.

Finally, given that most participants were non-native English speakers, we considered it important to test for potential differences in profile correlations between T1 and T2 assessments. Indeed, natives showed slightly higher average T1-T2 profile correlations ($q$ = .69) than non-natives ($q$ = .62). By implication, the $r_{TT}$ estimates provided here may thus underestimate reliability, providing a particularly conservative test. These estimates are slightly lower than the minimum recommendation for profile correlations of .70 based on a same-day retest interval [27], which is expected given our longer retest interval.

## HEXACO-100 domain and facet scales

As summarized in Table 1, domains had a median 13-day $r_{TT}$ of .88 (SD = .02, range = .86 to .92) versus a median α of .87 (SD = .02, range = .83 to .89). These αs, slightly lower than $r_{TT}$s for all but one domain (i.e., Agreeableness), were comparable to those from the comparison study by Lee and Ashton ([18]; $Mdn$ = .88, SD = .03, range = .82 to .89) that we report in Table 1, as well as a more recent meta-analysis ([2]; $Mdn$ = .84, SD = .02, range = .81 to .86). For facets, median $r_{TT}$ and α were .81 (SD = .04, range = .74 to .87) and .77 (SD = .07, range = .54 to .85), respectively. Our αs tended to be higher than those observed by Lee and Ashton ([18]; $Mdn$s = .70 and .73), although our rankings correlated highly with their student and online samples (ρs = .67 and .88; Table 2). In turn, facet $r_{TT}$s were higher than αs for all but three facets (mean difference $r_{TT}$−α: Δ = .04, median = .03, range = -.08 to .24). Though typically small, the disparities were more notable for facets with lower αs, as evidenced by a correlation of ρ = -.72 between α and Δ. For example, in the lowest quintile of αs (range = .54 to .70), the median difference between α and $r_{TT}$ was .14 (range = .06 to .24), whereas disparities were negligible in the highest quintile (α range = .82 to .85; Δ median = .0, range = -.08 to .01).

Details on inter-item correlations for domains, facets, and all items can be found in the Online Supplement.

**Table 2. Spearman correlations of $r_{TT}$, α, and $r_{CA}$ of HEXACO facets.**

| Estimate | α | $r_{TT}$ | $r_{CA}$ | LA $α_{Student}$ |
|---|---|---|---|---|
| $r_{TT}$ | .34 | | | |
| $r_{CA}$ | −.07 | .69 | | |
| LA $α_{Student}$ | .67 | .55 | .52 | |
| LA $α_{Online}$ | .88 | .36 | .07 | .69 |

α = alpha internal consistency estimates from our sample, averaged across both testing occasions. $r_{TT}$ = 13-day retest reliability. $r_{CA}$ = cross-rater agreement assessed by Lee & Ashton (2018) [18]. LA refers to the two samples in which Lee & Ashton (2018) calculated α, with appropriate subscripts.

## Relationships between $r_{TT}$, α, and the validity criterion $r_{CA}$ for HEXACO-100 facets

Table 2 reports Spearman correlations between $r_{TT}$s, αs, and $r_{CA}$s for the HEXACO-100 facets. While $r_{CA}$s correlated strongly ($ρ$ = .52) with only one of the three α values–$α_{Student}$, from the same sample–they demonstrated a stronger association with the $r_{TT}$ values from the present sample ($ρ$ = .69). Because $r_{TT}$ was also correlated with $α_{Student}$ ($ρ$ = .55), we conducted a *post-hoc* analysis to determine the partial associations among these three properties. When controlling for $r_{TT}$, the relationship between $α_{Student}$ and $r_{CA}$ was substantially attenuated ($ρ$ = .24). However, correcting the correlation between $r_{TT}$ and $r_{CA}$ for $α_{Student}$ only reduced the association to $ρ$ = .56.

## HEXACO-100 Items

Median 13-day $r_{TT}$ of the HEXACO-100 items was .65 (*M* = .65, *SD* = .08, range = .39 to .84). The average standard error of the correlations was .037 (range = .027 to .045). Estimates of $r_{TT}$s, standard deviations, and standard errors for all items of the HEXACO-100 can be found in S1 Table, with the five highest and lowest available in Table 3. Similar to the relationship found at the facet level, single-item $r_{TT}$ estimates correlated $ρ$ = .62 with their $r_{CA}$s. Interestingly, a *post-hoc* analysis found that item $r_{TT}$ and $r_{CA}$ were also highly correlated with the items' standard deviation: $ρ$s = .74 and .58, respectively.

**Table 3. HEXACO-100 items with highest and lowest test-retest reliabilities.**

| Item | Code | $r_{TT}$ |
|---|---|---|
| **Low** | | |
| I wouldn't pretend to like someone just to get that person to do favors for me | H1 | 0.39 |
| I wouldn't want people to treat me as though I were superior to them* | H4 | 0.46 |
| I don't allow my impulses to govern my behavior* | C4 | 0.47 |
| I generally accept people's faults without complaining about them* | A2 | 0.48 |
| I wouldn't use flattery to get a raise or promotion at work, even if I thought it would succeed | H1 | 0.48 |
| **High** | | |
| I find it boring to discuss philosophy | O4 | 0.84 |
| If I had the opportunity, I would like to attend a classical music concert | O1 | 0.83 |
| I would be quite bored by a visit to an art gallery | O1 | 0.80 |
| If I knew that I could never get caught, I would be willing to steal a million dollars (R) | H2 | 0.79 |
| I sometimes feel that I am a worthless person (R) | X1 | 0.77 |

Codes correspond to facet labels given in Table 1. $r_{TT}$ = 13-day test-retest reliability.

* indicates items not included in the HEXACO-60. (R) = reverse-keyed.

## HEXACO-60

Properties of the HEXACO-60 demonstrated similar patterns as the HEXACO-100, with $r_{TT}$ being slightly higher than α on average. Domains had a median $r_{TT}$ = .86 ($SD$ = .02, range = .82 to .89) and median α = .82 ($SD$ = .03, range = .80 to .87), with all αs ≤ $r_{TT}$. Median facet $r_{TT}$ was .76 ($SD$ = .05, range = .68 to .87) and median α was .72 ($SD$ = .08, range = .52 to .84). We again found the association between $r_{CA}$ and $r_{TT}$ for facets, $\rho$ = .57. This held for single items as well ($\rho$ = .67), for which the median 13-day $r_{TT}$ was .65 ($SD$ = .09, range = .39 to .84). As noted in Table 3, while three of the five lowest $r_{TT}$ items from the HEXACO-100 do not feature in the HEXACO-60, all five of the items with highest $r_{TT}$ are included in the shorter scale. A full breakdown of properties for HEXACO-60 domains, facets, and items can be found in the Online Supplement.

## Discussion

The HEXACO-PI-R, and particularly its two shorter versions, the HEXACO-100 and HEXACO-60, are currently among the most widely-used personality questionnaires. Surprisingly, however, the test-retest reliability ($r_{TT}$)–a key psychometric property of any psychological test–of their scales and items has not yet been systematically studied. Although several studies have reported HEXACO-PI-R stability estimates over intervals from a few months up to two years [7–9], we herein provided the first examination of *short-term* $r_{TT}$ for the English HEXACO-100 and HEXACO-60 to our knowledge. We found that $r_{TT}$s were slightly higher on average on both the domain and facet levels than internal consistencies (α), suggesting that the latter can underestimate scales' reliabilities, particularly at the lower end. $r_{TT}$s remained high even for those facets with low αs, which indicates that their items are likely reliable, just less intercorrelated than those of scales with a higher α. We also found evidence to suggest that α– but not $r_{TT}$–confounds reliability with non-random error sources of variance, indicated by strong associations between $r_{TT}$ and cross-rater agreement ($r_{CA}$), a common validity criterion, even when controlling for α.

The present study indicates that HEXACO-PI-R scales have comparable reliability as other established personality measures (see below). This is good news for researchers and practitioners, as the HEXACO-PI-R scales are freely available, meaning these measures can be applied without any costs that are associated with other proprietary personality measures. Moreover, our finding that the HEXACO-60 assesses HEXACO-PI-R domains, facets, and items with similar degrees of reliability to its longer 100-item variant supports its use in settings where a shorter version may be favored, such as clinical use or inclusion in a larger research project assessing many other variables. The finding that facet properties in the HEXACO-60 were consistent with the longer HEXACO-100, despite its being written only to measure domains, may suggest that researchers could consider interpreting facets when measuring the HEXACO domains with the shorter version, although this would need to be confirmed by predictive validity studies. In sum, our findings contribute further empirical backing for the use of HEXACO scales in research and practice.

### Striking similarities to other measures

The HEXACO-PI-R $r_{TT}$s observed here were remarkably similar to those reported for other popular personality scales, such as the Big Five, at all levels of the trait hierarchy. For example, McCrae and colleagues [13] found that median NEO-Personality Inventory-Revised (NEO-PI-R [30]) αs ranged from .71 to .77 in three samples, with an overall range of .50 to .87—compared to .54 to .85 in the present study. NEO-PI-R facet $r_{TT}$s also showed highly similar patterns to the present study, with median $r_{TT}$ = .82 and a slightly wider range of $r_{TT}$s = .72 to

.89 (where median facet $r_{TT}$ of the HEXACO-100 in our study was = .81, range = .74 to .87). McCrae and colleagues found consistent associations between $r_{CA}$ and $r_{TT}$, but none with αs after controlling for third variables, just as we observed in the present study.

Single items assessing the Big Five domains and facets also demonstrated similar levels of $r_{TT}$ to the present study. A recent study [16] reported median $r_{TT}$ of .64 *(M = .64, SD = .09*, range = .36 to .87) for the NEO-PI-R, whereas other research found median $r_{TT}$ of .66 (data for the *SD* and range were not available) for a 100-item IPIP scale [21, 25, 31]. The HEXACO-PI-R showed a similar $r_{TT}$ range as the NEO-PI-R, although the former was slightly narrower (i.e., *r* = .39 to .84 vs .36 to .87, respectively). The association between $r_{CA}$ and $r_{TT}$ has also been found for NEO-PI-R items ($\rho$ = .57, *p* < .001 [16]). These results suggest three things: (1) ever-accumulating evidence indicates that the "average" personality item is reliable at about $r_{TT}$ = .65; (2) items within contemporary scales vary substantially in quality, as indicated by several especially unreliable items; and (3) $r_{TT}$ is a good predictor of items' validity, making it useful quality criterion in scale development.

## Variation in item properties: Interpretations and implications

An inventory's reliability is a fundamentally desirable psychometric property, and a guiding theoretical claim of this manuscript has been that *retest* reliability in particular is a better index of reliability than other indices, particularly the more commonly-used Cronbach's α. We have also shown this empirically to be the case, with $r_{TT}$s > αs on average, and $r_{TT}$ consistently linked with validity criteria whereas α is not. But *why* should this be the case?

McCrae [14] offers one explanation, describing how to use the $r_{TT}$, α, and $r_{CA}$ of a given trait to parse its variance into common trait, method, and (items') specific variance components. In essence, the model postulates: Most items have unique valid variance [21, 32], and this unique variance is by definition *not* captured by α but *is* assessed by $r_{TT}$ (because α removes anything not common to all items); therefore, a trait scale that aggregates multiple items should have $r_{TT}$ > α. Our results support this model, with only three facet αs lower than $r_{TT}$s. In other words, most facet measures contain both information that is common to items written to measure the trait (e.g., Sincerity) and unique valid content specific to each item, ostensibly indexing a further personality *nuance* [14]. However, the ways that items vary in their individual $r_{TT}$s and contributions to higher-order trait αs is as of yet a relatively unexplored question.

**The role of item content.** As one starting point, we can compare how domains and facets in the HEXACO-100 differ in their $r_{TT}$s and αs. Of the top 10 most reliable items, Openness to Experience and Extraversion featured with three and four items, respectively. Conversely, Honesty-Humility and Agreeableness demonstrated the opposite pattern, containing three and four of the *least* reliable items. An examination of the least reliable items suggests that they may be especially dependent on the *contextuality* of the item, particularly when it describes a hypothetical "other." For example, responses to the least reliable item "I wouldn't pretend to like *someone* just to get that person to do favors for me" (emphasis added) may depend on a variety of circumstantial details the respondent imagines when responding: who is the "someone" and what relation do they have to me? Are they a friend, colleague, boss, or stranger? How important is the favor? The same participant could envision different situations with different individuals, stakes, and emotional investment when responding to the item on different testing occasions, ultimately leading to a low $r_{TT}$ on average. At face value, other low-$r_{TT}$ items seem to demonstrate similar levels of ambiguity (e.g., "I generally accept *people's* faults without complaining about them," "I wouldn't want *people* to treat me as though I were superior to them"; emphasis added) that leave the context unclear.

In contrast, the *most* reliable items are generally less vague in their contextual referents. "I find it boring to discuss philosophy," "If I had the opportunity, I would like to attend a classical music concert," and "I would be quite bored by a visit to an art gallery" are the three items with highest $r_{TT}$. While all three items assess Openness to Experience, more notable is that the content references specific situations that do not evoke the presence of a particular other individual. These Openness to Experience items leave little to the respondent's imagination.

Conversely, facets of Openness to Experience had the *lowest* α on average, with α for Unconventionality, Inquisitiveness, and Aesthetic Appreciation all falling below .70. They also had the greatest disparities between α and $r_{TT}$, with the latter at or near the median, and respective differences of .24, .14, and .14. This demonstrates two things. More generally, it serves as a reminder that a scale's actual reliability (as assessed by $r_{TT}$) is not well approximated by α. With regard to the Openness to Experience facets in particular, it may well be that the facet-level traits assessed are more abstract and therefore broader in content than facets of other domains such as, for example, Greed Avoidance from the Honesty-Humility domain–which has, in turn, one of the highest α values.

These speculations can and should be tested empirically. One way to explore questions of item content more generally is by recruiting a small pool of lay or expert raters (say, $N$ = 20–30 [33]) to assess the degree to which items differentially feature properties (including contextuality) that may be relevant for a variety of empirical criteria, such as $r_{TT}$ or $r_{CA}$. This would shed light on how the way items are written or the trait they are assessing is related to their empirical properties. Item ratings could also be averaged to generate scores for the broader facets they measure, which could then be compared to a rating of the same criterion but for the facet (or domain) alone. For example, how would a breadth rating for "Unconventionality" as a facet compare to the average of breadth ratings across its constituent items? Investigation of the nuanced ways in which measures of personality vary in content and at different levels of specificity could help generate new questions and hypotheses in future scale development.

Some work on these types of questions has already begun. Previous research [9] found that item variance strongly predicted items' $r_{CA}$ for both the NEO-PI-3 [34] and the full HEXACO-PI-R, as well as $r_{TT}$ for the latter, but not single-item internal reliability. $r_{TT}$ and $r_{CA}$ were also associated with item evaluativeness, observability, position, length, inclusion of a negation, and broad content domain (assigning Big Five and HEXACO domains to either "Engagement" or "Altruism"), although these factor loadings were more modest [9].

These are useful preliminary findings, but more properties still may be investigated to explore as of yet unexplained mechanisms driving items' validity. For example, the item properties studied by De Vries and colleagues [9] only achieved $R^2$ values of .06 and .17 for item variance, the strongest predictor of $r_{TT}$ and $r_{CA}$. Perhaps studying some of the aforementioned properties such as breadth, contextuality, and abstractness may clarify this issue; others have suggested additional candidates such as item ambiguity, complexity, and (social) importance [33]. Varying domains of content, such as the affective, behavioral, cognitive, and motivational components of items have also been suggested as possible candidates and thus preliminarily studied [35].

We would thus call for an integration of and extension to these varied lines of research. For example, while De Vries and colleagues [9] assessed $r_{TT}$ using data more appropriate for long-term stability estimates, follow-up work using the present findings could refine the relationships between $r_{TT}$ and item properties. Likewise, researchers could attempt to incorporate analyses of item content into studies that apply the variance decomposition techniques suggested by McCrae [14]. By drawing on the insights and resources from past work and across tests, we can iteratively make progress toward a comprehensive understanding of how items behave and why.

**Implications for scale development.** Taken together, the findings in the present study and subsequent examination of item properties may also offer personality researchers the potential to revise existing scales in an informed way, with the aim of keeping only those items that best measure the target traits. This, in turn, could result in survey administrations that capture more and better information with the same or even a smaller number of items, helping researchers to maximize the investment of their resources while also having more confidence in observed relationships with outcome variables of interest.

How exactly to assess the "goodness" of an item remains an open question, as evidenced by the previous section, but we would recommend that researchers begin by prioritizing items with higher $r_{TT}$s, standard deviations, and $r_{CA}$s, three highly intercorrelated empirical properties [9, 21, the present study]. We caution, though, that comparisons of these properties should only be used when choosing between items *for a single trait*, as it is conceivable that some traits may have reliability "ceilings" such that they are more difficult to assess than others–both for an individual about themselves or for an informant who knows them well–but this does not necessarily make them less of a trait. Further, the interpretation of these criteria is not always clear-cut; indeed, at least two criteria are logically intertwined: specifically, the stability of one's self-view ($r_{TT}$) is probably a necessary condition for an informant to agree with them ($r_{CA}$). Given the complexities of interpreting even apparently straightforward quality indices of items, let alone the complex interplay of factors at both the level of the written item and regarding the nature of the underlying trait itself, we are currently hesitant to offer concrete advice for survey generation. Instead, we encourage researchers to pursue questions–including those we pose above–that begin to paint a clearer picture of how items, traits, and their properties interact.

## Limitations and generalizability

A limitation of the present study is that we have only estimated $r_{TT}$ for the HEXACO-100, meaning half of the items that assess the HEXACO domains (as per the full HEXACO-PI-R) remain untested. Furthermore, both this study and much of the research that we have cited on $r_{TT}$ reports on relatively small samples. Measures for what arguably are the two most popular models of personality (HEXACO and Big Five) have $r_{TT}$ estimates based on samples $N < 500$, which rather pale in comparison to cross-sectional samples of (hundreds of) thousands of self- and informant-reports. This likewise limits the ability to generalize our findings, particularly given 1) the disproportionate number of non-native English speakers, and their lower overall consistency in responses compared to natives, which may have downwardly-biased our reliability estimates; and 2) that the sample was recruited using a paid online service, perhaps leading to a selection bias. To the former point, this may actually serve as encouraging, suggesting that the $r_{TT}$s reported here are actually lower limits for HEXACO-PI-R domain, facet, and item $r_{TT}$. To the latter, the αs observed in the present study are consistent with the substantially larger samples in recent studies and meta-analyses [2, 18], suggesting that our data were not irregular.

Finally, though we chose an approximately two-week retest interval primarily for empirical consistency with previous work, we found little robust theoretical rationale in the personality literature as to why to prefer two weeks to, say, one, three, four, six, or eight–aside from vague assumptions about true trait change and memory effects. As study of single-item properties in particular advances, researchers should investigate more thoroughly the differences in reliability across different retest intervals, while also integrating findings from the memory literature, to inform our understanding of appropriate retest intervals.

## Conclusions

Given (1) facet α estimates being generally lower than $r_{TT}$, (2) the unique, robust association between $r_{TT}$s and $r_{CA}$s, and (3) the remarkable replicability of these findings across multiple samples, questionnaires, and even models of personality, we reiterate the growing sentiment that $r_{TT}$ is a superior estimate of scale reliability to α (or its cousin ω). This is not a new position [11, 36], but it is one that has been overlooked given the convenience of estimating internal consistency. However, as participants become more readily available via online recruitment platforms, we see little reason to avoid–and much to be gained by–collecting $r_{TT}$ data as a standard procedure in scale development. We thus recommend calculation of $r_{TT}$ as routine practice in psychological research and argue that internal consistency should only be used to screen for data quality [13].

We conclude by advising researchers to pay special attention to item content when designing scales. The average personality item appears to be reliable at approximately $r_{TT}$ = .65. However, as shown in various samples, many items are still quite unreliable, ranging as low as the .30s. We suggest that the wide variability in $r_{TT}$ across personality scales is, in part, due to the inclusion of "poor" items, rather than solely indexing true variation in reliability across traits. We argue that $r_{TT}$ provides one straightforward means of identifying lower-quality items and replacing them with higher-quality ones [33, 37]. We also call for further investigation into the properties that are predictive of $r_{TT}$ and related validity criteria (e.g., $r_{CA}$, heritability, long-term stability). Ultimately, we encourage researchers to deliberately explore how components of item *content* relate to item *quality*, while at the same time considering the traits they intend to measure. We believe that devoting time and resources to these sorts of questions will move personality measurement–and thus our entire field–forward.

## Supporting information

**S1 Table. HEXACO-100 items and their descriptive statistics.** $r_{TT}$ = 13-day test-retest reliability. *SE* = standard error for the $r_{TT}$ estimate. *SD* = Standard deviation of the item. * indicates items not included in the HEXACO-60.
(DOCX)

**S1 File. Raw data for HEXACO-100 items, facets, and domains.**
(XLSX)

**S2 File. Facet and factor omegas for HEXACO-100 and HEXACO-60.**
(XLSX)

## Acknowledgments

We are very grateful to Kiboem Lee, Ph.D. and Michael C. Ashton, Ph.D. for providing us with item-level self- and informant-reports for the HEXACO-100.

## Author Contributions

**Conceptualization:** Sam Henry, René Mõttus.

**Data curation:** Sam Henry.

**Formal analysis:** Sam Henry.

**Funding acquisition:** Sam Henry.

**Methodology:** Sam Henry.

**Project administration:** Sam Henry.

**Supervision:** René Mõttus.

**Writing – original draft:** Sam Henry, Isabel Thielmann, Tom Booth, René Mõttus.

**Writing – review & editing:** Sam Henry, Isabel Thielmann, Tom Booth, René Mõttus.

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
