## [Decision Letter · Decision Letter 0]

1 Nov 2021

PONE-D-21-30871Test-retest reliability of the HEXACO-100PLOS ONE

Dear Dr. Henry,

Thank you for submitting your manuscript to PLOS ONE. We invite you to look at the reviewer's suggestions and think whether they could be used to improve the article. Please submit your revised manuscript by Dec 16 2021 11:59PM. If you will need more time than this to complete your revisions, please reply to this message or contact the journal office at plosone@plos.org. Please include the following items when submitting your revised manuscript:A rebuttal letter that responds to each point raised by the academic editor and reviewer(s). You should upload this letter as a separate file labeled 'Response to Reviewers'.A marked-up copy of your manuscript that highlights changes made to the original version. You should upload this as a separate file labeled 'Revised Manuscript with Track Changes'.An unmarked version of your revised paper without tracked changes. You should upload this as a separate file labeled 'Manuscript'.

We look forward to receiving your revised manuscript.

Kind regards,

Frantisek Sudzina

Academic Editor

PLOS ONE

Journal Requirements:

Reviewers' comments:

Reviewer's Responses to Questions

**Comments to the Author**

1. Is the manuscript technically sound, and do the data support the conclusions?

Reviewer #1: Yes

Reviewer #2: Yes

2. Has the statistical analysis been performed appropriately and rigorously? 

Reviewer #1: Yes

Reviewer #2: Yes

3. Have the authors made all data underlying the findings in their manuscript fully available?

Reviewer #1: Yes

Reviewer #2: Yes

4. Is the manuscript presented in an intelligible fashion and written in standard English?

Reviewer #1: Yes

Reviewer #2: Yes

5. Review Comments to the Author

Reviewer #1: Thank you for the opportunity to review this paper. I see it as being both interesting and very important to researchers using the HEXACO 60 and 100 personality inventories. I also really enjoyed the analysis of and discourse about the item-level data. I believe that this paper is likely to become the ‘go-to’ paper for people wishing to cite evidence of the reliability of these measures. Overall, I have only two relatively minor suggestions/thoughts.

1. If the authors have more information they can share about the sample (e.g., country of origin, education levels), I strongly encourage them to add this information to the Participants sections, if there is space. My thinking here is that these details may be important for future researchers who, for example, conduct a similar study but receive different results and wish to understand why.

2. After reading the third paragraph on page 13, which pondered the effects of items’ contextualisation levels on reliability, I wondered whether contextualisation levels would be predictably positively associated with rTT but negatively associated with α. For example, whether a person likes poetry _today_ is probably very strongly associated with whether they like it tomorrow, next week, next year, and so on (high rTT). But it’s not hard to imagine there would exist plenty of people who love poetry but are indifferent to, say, classical music or ancient ruins, thus the contextualised items contribute negatively to alpha. I concur with the authors’ speculation that more generic/less contextualised items (e.g., a hypothetical item, “I like artistic things”) may undermine rTT, for all the reasons the authors mentioned (e.g., what artistic things are they thinking about in that moment? Have they enjoyed/not enjoyed a recent artistic experience?). And I could see how such an item would positively influence alpha, as the generic item would represent, to some extent, any of the specific/contextualised items in the same scale. Anyway these are just thoughts; I do _not_ insist the authors should to include them in their revision.

Reviewer #2: Review PONE-D-21-30871: “Test-retest reliability of the HEXACO-100”

Many thanks for the opportunity to review this manuscript, which provides - once again - evidence that alpha reliability is a less optimal parameter than test-retest reliability. Based on my reading of the manuscript, I have a few suggestions and comments:

1. One of the open questions for me is what the most optimal time period is to establish test-retest reliability. The authors chose 12 days (please provide mean and SD of number of days or even hours between the two ratings; and please check whether the individual number of days has an effect on r(tt)!), but I’m not sure whether this is the optimal time period and what is actually the most optimal time period for personality questionnaires. That is, in the introduction and the discussion, I would like the authors to explain a bit more, based maybe on memory research (which, of course, also shows large individual differences) and based on the traitedness of a construct and the possible time frame for changes to occur, what kind of time frame would be most optimal to establish r(tt).

2. With respect to the above time frame, the findings can also be used to comment on McCrae’s (2015) approach to distinguish trait, method, specific, and error variance components. McCrae notes that specific variance is obtained by subtracting alpha from r(tt), but in most cases this would yield a negative specific variance in the current study. As McCrae notes: “By definition, [...] specific variance in an item is not shared by other items in the scale, so it detracts from alpha. However, in retest designs, the same items, with the same specific variance, are readministered, and they may elicit the same response. Item-specific variance could thus account for the fact that retest reliability is greater than alpha, especially if we also assume that method variance is stable over short intervals.” (McCrae, 2015, p. 2)

That is, McCrae’s formula implies that the time period between two measures of the same construct should depend on the specific variance (i.e., if there is more specific variance, the time period should be longer, because else r(tt) is bound to be greater than alpha. I’d love the authors to comment on this. Note: I must admit there are notable problems with McCrae’s approach, something that is long overdue being commented on.

3. I wondered about the criteria to establish whether an item is a ‘good’ item. One could argue that both r(ca) and r(tt) are important, and not just r(ca). But how to weigh these is - to me - an open question. Logically, r(tt) is a necessary, but not sufficient, condition for r(ca) (i.e., a highly temporally stable item may not be observable, and thus have a low r(ca), whereas r(ca) may be a sufficient condition for r(tt) (if items are really observable and there is high r(ca), by necessity there is a high r(tt)). But the question is whether you only want to have observability criteria (or other criteria aligned with r(ca), e.g., ‘item domain’, see De Vries et al., 2016) properties in a personality questionnaire. I would love to see the authors make a statement about this in the discussion and maybe even suggest which (24? 48?) items would provide the most suitable short measure of the HEXACO-100 (with coverage of each facet) according to their criteria.

4. Last but not least, I would love the authors to make the title a bit more informative about the implications of the manuscript, especially with respect to the importance of test-retest reliability and the fact that alpha reliability should be less often used as a measure of reliability. As a final note, please refrain from using the term ‘internal consistency’ and/or explain that it is a misnomer, because alpha does not measure internal consistency (with thousands of items, any scale has a high alpha, but can have practically zero internally consistency). See Sijtsma (2009); just call it ‘alpha reliability’ or ‘internal reliability’.

McCrae, R. R. (2015). A more nuanced view of reliability: Specificity in the trait hierarchy. Personality and Social Psychology Review, 19(2), 97-112.

6. PLOS authors have the option to publish the peer review history of their article (what does this mean?). If published, this will include your full peer review and any attached files.

Reviewer #1: No

Reviewer #2: **Yes: **Reinout E. de Vries

---

## [Author Response · Author response to Decision Letter 0]

22 Dec 2021

Response to Reviewers | PONE-D-21-30871R1 | Test-Retest Reliability of the HEXACO-100

Journal Requirements:

We have carefully reviewed PLOS ONE’s style requirements and believe that all files meet these requirements. 

The minimal underlying dataset for test-retest reliability has now been submitted as Supporting Information (S2_File.xlsx) and can also be found in the project page on the Open Science Framework website, listed in the manuscript (https://osf.io/wz3du/). We also include the Lee & Ashton (2018) article, which provides facet alphas and cross-rater agreement estimates in Table 4, as is now noted in the Data Availability statement.

We now also note in the Data Availability statement how to access the single-item cross-rater agreement data: “Though the authors of Lee & Ashton (2018) do not specifically indicate in their article how to access these data, we simply sent them an email requesting the raw self-/observer data and explaining how we intended to use them. We would thus confirm that, to our knowledge, others can access these datasets and would be able to access these data in the same manner as the authors of the present manuscript by contacting the authors Lee & Ashton (2018) in a similar manner.”

See previous response; our data is made available at the following data repository: https://osf.io/wz3du/.

We now include a full ethics statement in the “Methods” section that includes the full name of the ethics committee at the University of Edinburgh who approved our study. We also note that consent was in written form, given online at the time of completing the survey.

We have revised the reference section and can confirm that 1) all references are up to date and correct, and 2) none of our references have been retracted.

----------- 

Reviewer #1: Thank you for the opportunity to review this paper. I see it as being both interesting and very important to researchers using the HEXACO 60 and 100 personality inventories. I also really enjoyed the analysis of and discourse about the item-level data. I believe that this paper is likely to become the ‘go-to’ paper for people wishing to cite evidence of the reliability of these measures. Overall, I have only two relatively minor suggestions/thoughts.

Thank you for this very positive evaluation of the manuscript.

1. If the authors have more information they can share about the sample (e.g., country of origin, education levels), I strongly encourage them to add this information to the Participants sections, if there is space. My thinking here is that these details may be important for future researchers who, for example, conduct a similar study but receive different results and wish to understand why.

Though we did not explicitly ask participants about these variables, we were able to access information via Prolific Academic for many of our participants’ first language, country of birth and residence, student status, and occupational status. All of these are now included in the “Participants” subsection of our “Materials and Methods” section (lines 177-189). We did not have full data for every demographic variable because, at the time of extracting it, some participants had removed their information. Where we have a substantial amount of missing data (e.g., for student and occupational status) we mention this specifically. 

Reporting these demographic variables also allowed us to examine the effect of having English as a first language on rTT, as nearly three quarters of our sample were non-natives. There did indeed appear to be a slight difference between natives and non-natives. We now discuss this and its implications in the Methods, Results, and Limitations sections (lines 177-183, 246-252, 486-493), noting that our reported rTTs may thus be lower bound estimates.

2. After reading the third paragraph on page 13, which pondered the effects of items’ contextualisation levels on reliability, I wondered whether contextualisation levels would be predictably positively associated with rTT but negatively associated with α. For example, whether a person likes poetry _today_ is probably very strongly associated with whether they like it tomorrow, next week, next year, and so on (high rTT). But it’s not hard to imagine there would exist plenty of people who love poetry but are indifferent to, say, classical music or ancient ruins, thus the contextualised items contribute negatively to alpha. I concur with the authors’ speculation that more generic/less contextualised items (e.g., a hypothetical item, “I like artistic things”) may undermine rTT, for all the reasons the authors mentioned (e.g., what artistic things are they thinking about in that moment? Have they enjoyed/not enjoyed a recent artistic experience?). And I could see how such an item would positively influence alpha, as the generic item would represent, to some extent, any of the specific/contextualised items in the same scale. Anyway these are just thoughts; I do _not_ insist the authors should to include them in their revision.

We think that this is an interesting proposition and spent some time trying to fit it into the Discussion; we even went so far as to attempt to test this hypothesis (see below). Ultimately, we chose not to include it in the final document as we felt it distracted somewhat from the main purpose, which was simply to speculate on the wide variety of causes of variation in item properties. However, we agree that this would be worth exploring further in any work that specifically looks at contextuality. 

Excised: 

“Specifically, we examined the correlations of item rTT with both �s and rTTs of the facets and domains the items measure. For facets, we found a strong relation (ρ = .58) between single-item rTT and facet rTT on the one hand (which is reasonable given that each item effectively contributes 25% to their facet’s rTT). On the other hand, single-item rTT and facet �s had a much smaller association, and in the opposite direction than we might have expected suggested in the previous paragraph (ρ = .11). When expanding these analyses to domains, these relations effectively went to zero. The Spearman’s correlation between item and domain rTT was ρ = .10 and between item rTT and domain �, ρ = -.01. Based on this, there is not strong evidence to suggest that, were it the case that more specific, less contextually variable items are more reliable than more general items, that has to come at the “cost” of lower �.”

----------- 

Reviewer #2: Review PONE-D-21-30871: “Test-retest reliability of the HEXACO-100”

Many thanks for the opportunity to review this manuscript, which provides - once again - evidence that alpha reliability is a less optimal parameter than test-retest reliability. Based on my reading of the manuscript, I have a few suggestions and comments:

1. One of the open questions for me is what the most optimal time period is to establish test-retest reliability. The authors chose 12 days (please provide mean and SD of number of days or even hours between the two ratings; and please check whether the individual number of days has an effect on r(tt)!), but I’m not sure whether this is the optimal time period and what is actually the most optimal time period for personality questionnaires. That is, in the introduction and the discussion, I would like the authors to explain a bit more, based maybe on memory research (which, of course, also shows large individual differences) and based on the traitedness of a construct and the possible time frame for changes to occur, what kind of time frame would be most optimal to establish r(tt).

We are happy to elaborate on these issues. We have added a paragraph in the Methods section with a comprehensive description of times: both the days, hours, and minutes between survey administration as well as the time it took participants to complete the survey at each time point. We then examined whether either of these had an impact on retest reliability by correlating them with participants T1-T2 overall profile consistency. In summary, we found no evidence that interval length or overall duration had an effect on reliability, but please see lines 228-245 for more details.

Regarding the second point, we were not able to find much information on ideal intervals aside from previous empirical work comparing different intervals, which we now include in the manuscript – both in the Introduction and the Discussion. We now more explicitly note that we chose our interval (actually closer to 13 than 12 days) largely to be consistent with and comparable to previous research. We also dedicate more space to previous research which has discussed the issue of appropriate interval length and go on to note that there is little theoretical rationale even in the typically-cited papers that refers to memory work in particular. 

We thus agree that these are interesting and important questions that need to be explored with respect to retest reliability, especially in an era where single-item properties are receiving more and more attention (see lines 119-129 and 495-501 where we comment on this). 

2. With respect to the above time frame, the findings can also be used to comment on McCrae’s (2015) approach to distinguish trait, method, specific, and error variance components. McCrae notes that specific variance is obtained by subtracting alpha from r(tt), but in most cases this would yield a negative specific variance in the current study. As McCrae notes: “By definition, [...] specific variance in an item is not shared by other items in the scale, so it detracts from alpha. However, in retest designs, the same items, with the same specific variance, are readministered, and they may elicit the same response. Item-specific variance could thus account for the fact that retest reliability is greater than alpha, especially if we also assume that method variance is stable over short intervals.” (McCrae, 2015, p. 2)

That is, McCrae’s formula implies that the time period between two measures of the same construct should depend on the specific variance (i.e., if there is more specific variance, the time period should be longer, because else r(tt) is bound to be greater than alpha. I’d love the authors to comment on this. Note: I must admit there are notable problems with McCrae’s approach, something that is long overdue being commented on.

Another excellent point. Several of the authors on this manuscript are actually also interested in re-visiting this technique and have plans to do so in upcoming projects, utilizing some of the ideas discussed in this paper. That is to say, we wholeheartedly agree that this needs to be commented on in more detail, although we recognize that a full discussion of McCrae’s method is probably beyond the scope of the present manuscript.

To the first point, we have now included a few paragraphs commenting on this in the Discussion (lines 373-382). Just as a note, we wonder whether the reviewer may have mis-read Table 1, because for only 3 facets is � > rTT, which is what McCrae’s approach would predict. For example, we now include the following: “Most items have unique valid variance [21,32], and this unique variance is by definition not captured by � but is assessed by rTT (because � removes anything not common to all items); therefore, a trait scale that aggregates multiple items should have rTT > �. Our results support this model, with only three facet �s lower than rTTs. In other words, most facet measures contain both information that is common to items written to measure the trait (e.g., Sincerity) and unique valid content specific to each item, ostensibly indexing a further personality nuance [32]” (lines 378-384).

3. I wondered about the criteria to establish whether an item is a ‘good’ item. One could argue that both r(ca) and r(tt) are important, and not just r(ca). But how to weigh these is - to me - an open question. Logically, r(tt) is a necessary, but not sufficient, condition for r(ca) (i.e., a highly temporally stable item may not be observable, and thus have a low r(ca), whereas r(ca) may be a sufficient condition for r(tt) (if items are really observable and there is high r(ca), by necessity there is a high r(tt)). But the question is whether you only want to have observability criteria (or other criteria aligned with r(ca), e.g., ‘item domain’, see De Vries et al., 2016) properties in a personality questionnaire. I would love to see the authors make a statement about this in the discussion and maybe even suggest which (24? 48?) items would provide the most suitable short measure of the HEXACO-100 (with coverage of each facet) according to their criteria.

We have also been pondering how best to operationalize the “goodness” of an item lately, and we have yet to come to any clear conclusion. We agree with the speculations here, but (as we now mention in the Discussion) are relatively hesitant to suggest a specific subset of items given how much there is yet to be understood about the interplay of item properties (content, empirical quality, and otherwise). 

We offer some tentative starting points for prioritising one item for a given trait over another (e.g., selecting items with high variance, rTT, and rCA) but do not feel we have enough evidence to propose a specific subset of HEXACO items for future research. See the section “Implications for scale development” from line 454 for our full commentary on the topic. 

4. Last but not least, I would love the authors to make the title a bit more informative about the implications of the manuscript, especially with respect to the importance of test-retest reliability and the fact that alpha reliability should be less often used as a measure of reliability. As a final note, please refrain from using the term ‘internal consistency’ and/or explain that it is a misnomer, because alpha does not measure internal consistency (with thousands of items, any scale has a high alpha, but can have practically zero internally consistency). See Sijtsma (2009); just call it ‘alpha reliability’ or ‘internal reliability’.

Thank you for this suggestion. We agree and have now changed the title to be “Test-Retest Reliability of the HEXACO-100 – and the Value of Multiple Measurements for Assessing Reliability”

Regarding the point about alpha, we went back and forth between the two recommendations and have ultimately chosen to continue referring to it as “internal consistency” for ease of differentiating it from rTT throughout the paper and to be consistent with the literature (we, e.g., use a direct quote from McCrae who also referred to internal consistency for alpha). That said, we now include a clarification on the first page that addresses the Reviewer’s point and explains that using “internal consistency” to describe alpha is a misnomer: “We note that alpha does not measure internal consistency of items per se: with hundreds or thousands of items, any scale has high alpha but can have practically zero consistency among individual items. Instead, a scale’s alpha indexes the expected consistency among hypothetical item aggregates containing the same number of items as the scale. Nonetheless, in line with common usage and to clearly distinguish it from rTT, we will also refer to alpha as a measure of internal consistency” (lines 79-84).

---

## [Editor Report · Decision Letter 1]

26 Dec 2021

Test-retest reliability of the HEXACO-100 - and the value of multiple measurements for assessing reliability

PONE-D-21-30871R1

Dear Dr. Henry,

We’re pleased to inform you that your manuscript has been judged scientifically suitable for publication and will be formally accepted for publication once it meets all outstanding technical requirements.

Kind regards,

Frantisek Sudzina

Academic Editor

PLOS ONE
---

## [Editor Report · Acceptance letter]

4 Jan 2022

PONE-D-21-30871R1 

Test-Retest reliability of the HEXACO-100 – and the value of multiple measurements for assessing reliability 

Dear Dr. Henry:

I'm pleased to inform you that your manuscript has been deemed suitable for publication in PLOS ONE. Congratulations! Your manuscript is now with our production department. 

Kind regards, 

on behalf of

Dr. Frantisek Sudzina 

Academic Editor

PLOS ONE